# Transition metal catalysis in the mitochondria of living cells

María Tomás-Gamasa[1], Miguel Martínez-Calvo[1], José R. Couceiro[1] & José L. Mascareñas[1]

The development of transition metal catalysts capable of promoting non-natural transformations within living cells can open significant new avenues in chemical and cell biology. Unfortunately, the complexity of the cell makes it extremely difficult to translate standard organometallic chemistry to living environments. Therefore, progress in this field has been very slow, and many challenges, including the possibility of localizing active metal catalysts into specific subcellular sites or organelles, remain to be addressed. Herein, we report a designed ruthenium complex that accumulates preferentially inside the mitochondria of mammalian cells, while keeping its ability to react with exogenous substrates in a bioorthogonal way. Importantly, we show that the subcellular catalytic activity can be used for the confined release of fluorophores, and even allows selective functional alterations in the mitochondria by the localized transformation of inert precursors into uncouplers of the membrane potential.

[1] Centro Singular de Investigación en Química Biolóxica e Materiais Moleculares (CIQUS), Departamento de Química Orgánica, Universidade de Santiago de Compostela, Santiago de Compostela 15782, Spain. Correspondence and requests for materials should be addressed to J.L.M. (email: joseluis.mascarenas@usc.es).

The functioning of the cell depends on the regulated action of thousands of different enzymes that have evolved to catalyse a wide range of chemical reactions. In many cases, the correct working of these enzymes requires an appropriate localization in specific organelles and/or subcellular sites[1]. This is the case, for instance, for mitochondrial enzymes, which need to be associated with different mitochondrial components in order to exert their critical role in cellular respiration[2–4].

Given the biological relevance of this type of intracellular localization, it is reasonable to envision that installing artificial enzymes with non-natural functions in designed cellular compartments might unveil new opportunities for probing and manipulating cell biology. While recent years have witnessed notable advances in the implementation of evolved enzymes capable of achieving *in vitro* non-natural transformations[5–7], including artificial metalloenzymes[8–12], engineering of this type of systems in *in vivo* settings is far from obvious.

An alternative and highly appealing way to generate localized, abiotic catalytic activities inside cells could be based on the targeted subcellular delivery of transition metal catalysts. However, achieving catalytic organometallic reactions inside living cells is not trivial, and many problems associated to the activity, stability, aqueous and biological compatibility, orthogonality, and cell entrance can be envisioned. The living cell is a very complex, compartmentalized and dynamic entity, with a very high concentration of biomolecules, ions and other structures in complex equilibrium, and can therefore be considered as a very stringent reaction medium.

Despite all these potential complications, recent data suggest that certain transition metal derivatives can promote intracellular reactions through typical organometallic mechanisms. Particularly relevant in this context has been the pioneering work by Meggers and coworkers, who demonstrated that discrete organoruthenium complexes could be used for the *in cellulo* uncaging of allylcarbamate protected (alloc) amines[13,14]. Our laboratory has reported that this type of catalysts can be employed for the uncaging of DNA binders[15]. Importantly, while these results point to intracellular reactions, a recent publication by Waymouth and Wender suggests that, at least in 4T1 cells, these Ru complexes are readily washed out with PBS, and raises doubts on the intracellularity of the metal catalysis[16].

Other important contributions in the area of *in cellulo* metal catalysis deal with the use of palladium complexes, albeit success in these transformations seems to require the use of heterogeneous nanostructured palladium species, and in most of the cases, *in cellulo* imaging of the catalytic responses has been analysed after fixation of the cells[17–19].

All these data confirm that achieving organometallic catalytic reactions of exogenous substrates within living cells is certainly tricky[20–25]. While the field is in its infancy and further progress requires the development of new biocompatible transformations, there is an urgent need to make operative catalysts that are well retained inside cells and ensure intracellular activities. In addition, there are many other questions that remain to be addressed. Is it possible to concentrate the catalyst within a specific organelle/environment while keeping its activity, and without generating toxicity? Would it be possible to visualize the catalyst within the cell and the organelles? Is it possible to use the confined catalyst to generate a differentiated biological effect? Could this localization provide a functional advantage?

In the present manuscript, we provide some answers to these questions by describing a designed ruthenium conjugate that is capable of accumulating in the mitochondria of living cells and promoting a localized uncaging of alloc/allyl protected exogenous substrates, reaction that operates through typical pi-allyl-mediated mechanisms (Fig. 1a,b). In addition to

confirming the intracellular performance of the catalyst, this work represents the first demonstration that it is possible to generate abiotic, bioorthogonal catalytic power in subcellular compartments of living cells.

## Results

**Mitochondria targeting**. Mitochondria are complex organelles found in almost all eukaryotic cells that play vital roles in the regulation of cellular function and survival[4,26]. Therefore, along the last years, there have been many efforts to engineer cell permeable vectors capable of targeting this organelle[27–29]. A particularly successful delivery platform is based on the use of triphenylphosphonium (TPP) cations, which exhibit a single positive charge resonance-stabilized over three phenyl groups and a large hydrophobic surface area. It is known that the accumulation of these type of lipophilic cationic compounds is driven by the membrane potential across the mitochondrial inner membrane[30]. TPP cations can be therefore taken up by the mitochondrial matrix, where they are freely soluble or adsorbed on the leaflet of the inner membrane depending on the hydrophobicity of the cargo[30,31].

Even though the functional complexity of mitochondria suggests that it might be incompatible with organometallic chemistry, we were intrigued by the possibility of accumulating active transition metal complexes in this organelle using a TPP delivery vector. Following recent work on the use of discrete ruthenium catalysts to promote an orthogonal removal of alloc protecting groups[13–15], we decided to explore the use of 2-quinolinecarboxylate (QC) ligand as platform for the introduction of the required phosphonium-targeting groups[14,32]. Towards this end, we prepared the complex **RuL1** (Fig. 2), in which the metallic centre contains a modified QC ligand linked to a TPP group through a relatively hydrophobic alkyl chain. This new structure is a lipophilic dication that might have an intrinsic preference for accumulation in the mitochondria. The hydrophobic linker should counterbalance the Born energy of the two charges, and favour the membrane crossing processes. This complex was prepared in four steps from 4-bromoquinoline-

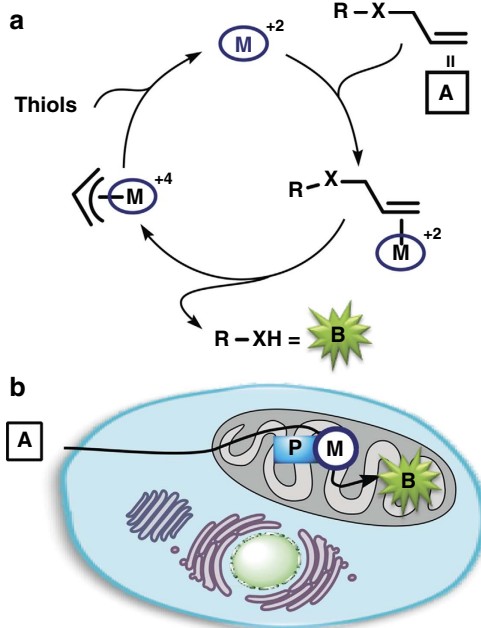

**Figure 1 | Aim of the present study.** (**a**) Transition metal pi-allyl chemistry. (**b**) Preferential accumulation of the metal catalyst in the mitochondria owing to the targeting group P. Ⓜ, metal complex; Ⓟ, mitochondria-targeting group.

**Figure 2 | Ruthenium complexes Ru1 and RuL1.** Synthesis of RuL1: (a) **1** (1.0 equiv.), allyl bromide (1.6 equiv.), KHCO$_3$ (1.7 equiv.), DMF, 12 h, 50 °C, 75%; (b) 3-N-methyl-propanol (5.0 equiv.), DMSO, 1 h, 100 °C, 87%; (c) (3-carboxypropyl)triphenylphosphonium bromide (1.0 equiv.), EDC (2.5 equiv.), DMAP (2.0 equiv.), 12 h, room temperature, 30%; (d) [RuCp(CH$_3$CN)$_3$]PF$_6$ (1.0 equiv.), DCM, 12 h, room temperature, 65%. DCM, dichloromethane; DMAP, 4-dimethylaminopyridine; DMF, N,N-dimethylformamide; EDC, 1-ethyl-3-(3-dimethylaminopropyl)carbodiimide.

2-carboxylic acid (**1**)[33], via the sequence of steps shown in the Fig. 2, which involves the preparation of the ligand and its complexation to the ruthenium precursor [RuCp(CH$_3$CN)$_3$]PF$_6$ (see Supplementary Methods and Supplementary Figs 1–4 for details).

For comparison purposes, we also made the parent complex **Ru1**, which lacks the TPP-containing appendage. Although previous studies had indicated that the catalytic activity of these ruthenium derivatives is favoured by the presence of electron donor groups in the quinoline, in the context of our research these derivatives, which demand a more elaborated synthesis, are not required[14].

**In vitro and cellular reactivity**. To evaluate the catalytic capabilities of these complexes, we selected the transformation of the bis-allyloxycarbonyl-protected rhodamine (**Rho-Alloc**) into its corresponding free-amine derivative (**Rho**), reaction that can be monitored by fluorescence (Fig. 3a)[13–15].

Before testing the reactivity in living cells, we analysed the in vitro performance of both metal complexes using biologically relevant reaction media. The experiments were carried out using 5% of the ruthenium complex either in water, PBS (pH 7.2) or HeLa cell lysates, and a concentration of **Rho-alloc** of 100 μM. As it can be observed in Fig. 3a,b, the reaction rates and conversions depend on the reaction conditions. Both complexes presented activity, although **RuL1** was much more active than complex **Ru1** in all the media studied, even in cell lysates. In relative terms, the activity in water and PBS was always higher for both catalysts, with a substantial decrease in cell lysates. However, **RuL1** led to a lower decrease than **Ru1**. Nevertheless, in both cases, rate and turnover were lower than those previously described for **Ru1** using other substrates and reaction media, in part because of solubility problems of the probe (Supplementary Table 1)[14].

Even though the reactivity was poor in the above conditions, we examined their performance in living cells using human cervical cancer cells (HeLa cells). Since our objective is the installation of artificial catalysts in a subcellular site, we carried out the experiments in an inverse way to that used in previous studies, in which the substrate was added to the cells before the catalyst[13–15]. To avoid misinterpretations arising from potential toxic effects of the metal complexes, we first analysed the cell viability. Gratifyingly, using the MTT (3-(4,5-dimethylthiazol-2-yl)-2,5-diphenyltetrazolium bromide) assay[34], we were glad to observe that both **Ru1** and **RuL1** are quite innocuous, and only

promote a decrease in the viability at concentrations >100 μM (30% decrease) and after 24 h of incubation (after 2 h we did not detect any toxicity, Supplementary Fig. 5).

Considering the previously mentioned recent report suggesting that ruthenium catalysts such as **Ru1** are readily washed out with PBS in cellular settings[16], we performed parallel experiments to compare the catalytic performances of **Ru1** and **RuL1** (Fig. 3; Supplementary Fig. 6). Therefore, cultured cells were treated with either ruthenium complex **Ru1** or **RuL1** (25 μM) for 15 min in fresh DMEM-FBS (DMEM supplemented with 5% of fetal bovine serum), followed by washing for 20 min with PBS[16]. The resulting cells were then incubated with **Rho-alloc** (100 μM) in fresh DMEM-FBS for 30 min and washed for 4 min with PBS. In consonance with that observed by Waymouth and Wender[16], the green staining inside cells treated with **Ru1** was very weak (Fig. 3d, panel B). In contrast, and importantly, we observed an intense green fluorescence in cells treated with **RuL1** (Fig. 3d, panel C). This suggests that the presence of the TPP appendage allows a much better intracellular retention of the complex (**RuL1**). It is important to note that control experiments with **Ru1** revealed that if the cells are washed with DMEM-FBS instead of PBS there is clear intracellular fluorescence (Supplementary Fig. 7).

Interestingly, when the above uncaging of **Rho-alloc** with **RuL1** (100 μM) was carried out in cells previously treated with the mitochondrial dye tetramethylrhodamine ethyl ester (TMRE, 100 μM)[35], we observed that the intracellular green staining co-localizes, at least in part, with the TMRE dye (Fig. 3e, panels D–F). This was better appreciated by comparing this costaining with that obtained in cells treated with **Ru1**, which promoted a less intense and diffuse fluorescence signal extended across the cytoplasm (Supplementary Fig. 7). Control experiments in the absence of ruthenium complexes confirmed that there is not an observable intracellular fluorescence even after several hours of incubation with the caged fluorophore.

These results confirmed that the structure of the catalyst influences the reactivity profile, and that **RuL1** generates a more intense fluorescence inside cells, mainly localized in the mitochondria surroundings. Inductively coupled plasma mass spectrometry (ICP-MS) analysis of cells incubated with equal concentrations of both complexes further confirmed that **RuL1** is better retained by cells, and more concentrated in the mitochondria inside the cells, with respect to **Ru1** (Supplementary Table 2).

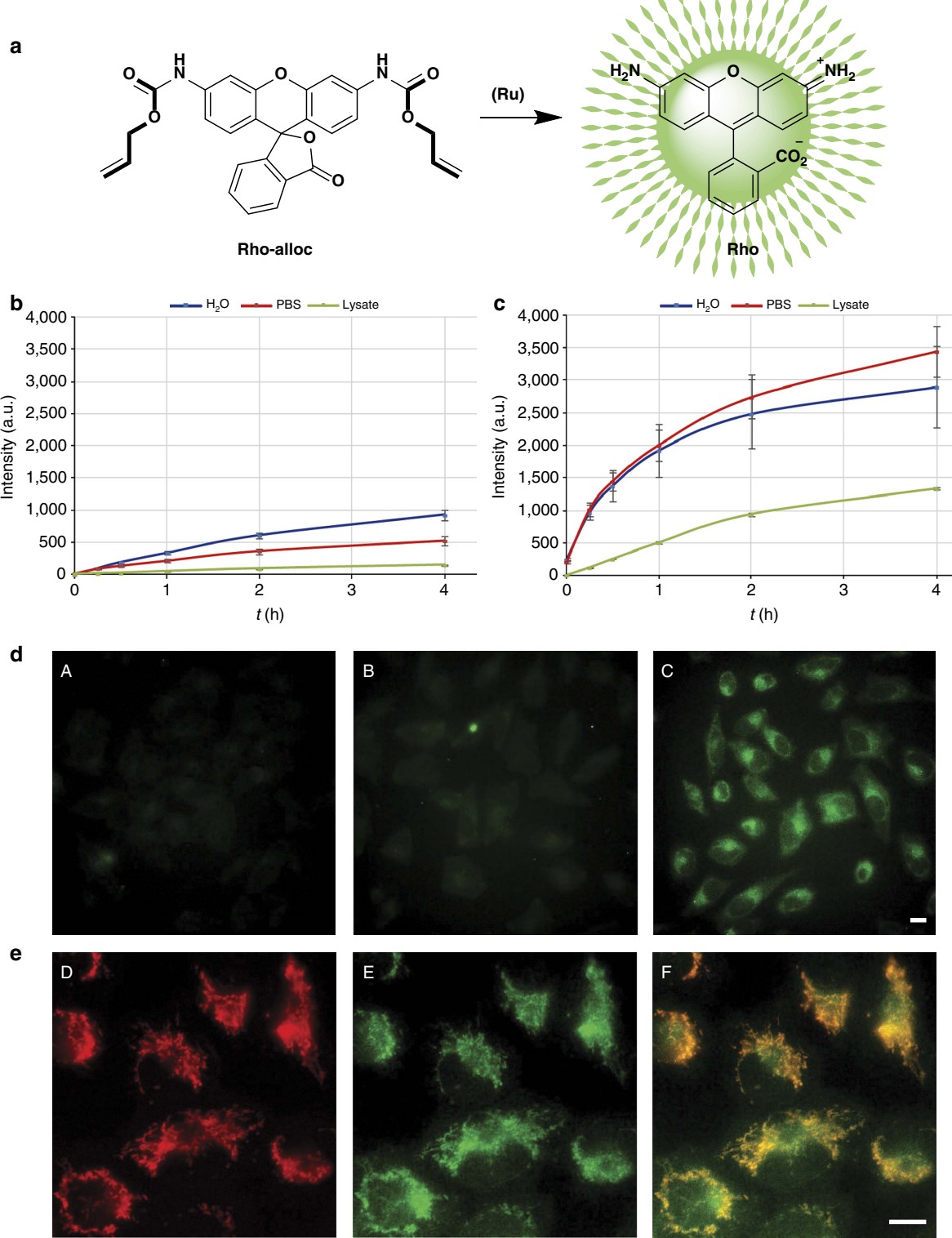

**Figure 3 | Uncaging experiments of Rho-alloc with Ru1 and RuL1. (a)** Schematic representation of the uncaging reaction of **Rho-alloc. (b,c) Ru1-** (**b**) and **RuL1**-(**c**) catalysed uncaging of **Rho-alloc** under biologically relevant conditions. Reaction conditions: **Rho-alloc** (500 μM, 1 equiv.), ruthenium complex (5 μM, 0.05 equiv.) in 120 μl solution of either HeLa cells lysates, PBS ( + 5 mM of glutathione, 10 equiv.) or water ( + 5 mM of glutathione, 10 equiv.). (**d**) Micrographs showing the fluorescence of the product (**Rho**) in cells treated with the ruthenium complexes (25 μM) for 15 min (except in case A), washed for 20 min with PBS; treated with **Rho-alloc** (100 μM) for 30 min and washed again for 4 min with PBS. (A) control experiment, not ruthenium complex added; (B) results with complex **Ru1**; (C) results with complex **RuL1**. (**e**) Fluorescent micrographies of HeLa cells after **RuL1**-catalysed uncaging of **Rho-alloc**. (D) Mitochondrial labelling with TMRE (red channel)[35]. (E) Fluorescence resulting from the **RuL1** promoted reaction (green channel) and (F) merging of D and E. Reaction conditions: cells were incubated with **RuL1** (50 μM) and TMRE (100 nM) for 30 min, followed by double washing with DMEM-FBS and incubation with **Rho-alloc** (100 μM) for 30 min. Scale bar, 12.5 μm. Error bars represent the standard deviation of measurements in three different samples.

To gain more information on the system, and to track the intracellular presence of the metal complexes, we pursued the development of derivatives equipped with fluorescent labels. Previous approaches to make fluorescent TPP cations have been based on the attachment of external fluorophores[36,37]. However, this strategy introduces important structural changes in the probe. We envisioned that an alternative based on the replacement of one of the phenyl rings of the phosphonium group by a pyrene derivative might allow fluorescent monitoring while keeping the required hydrophobicity/charge balance of the targeting vector. Thus, we prepared the pyrene-phosphonium complex **RuL2**, as well as the analogue **RuL3** that lacks the phosphonium group and could therefore be used as reference (Fig. 4a; see the Supplementary Methods for synthetic details). The ultraviolet–visible spectra of **RuL2** and **RuL3** showed a set of bands of the QC unit centred at ca. 275 nm, whereas at lower energy there is the typical set of bands corresponding to the pyrene ring with a $\lambda_{max}$ at 345 nm. The fluorescence spectra displayed maxima at 385 and 390 nm for **RuL2** and **RuL3**, respectively (Supplementary Fig. 8).

Gratifyingly, the addition of TMRE (100 nM) and **RuL2** (50 µM) to HeLa cells, followed by double washing with DMEM-FBS elicited a clear intracellular blue fluorescence after 5 min of incubation (Fig. 4b, panel B), staining that presents an excellent overlay (Mander's coefficient $\sim$90%)[38] with the marker TMRE (Fig. 4b, panel C). Subsequent experiments revealed that a final catalyst concentration as low as 10 µM is even enough for the uptake and imaging. Importantly, cells treated with the conjugate **RuL3**, a control complex lacking the phosphonium moiety, showed a different staining pattern (Fig. 4c, panel B), as can be observed in the TMRE co-localization picture (Fig. 4c, panel C), with the fluorescence preferentially located in the perinuclear region.

Importantly, ICP experiments of cells incubated with the same concentrations of both complexes corroborated that the phosphonium-pyrene derivative **RuL2** was better retained than **RuL3**. Moreover, ICP analysis of the ruthenium content in isolated mitochondria of the cells, revealed a much higher proportion in the cells treated with **RuL2**. Therefore, the presence of the pyrene moiety in **RuL2** increased its cellular and mitochondrial retention with respect to the triphenylphosphonium analogue **RuL1** (Supplementary Table 2), which suggests that the phosphonium-pyrene unit is an excellent mitochondrial tagging fluorescent group, and might be used for other applications[39].

While the appearance and morphology of the cells suggested that both ruthenium complexes are innocuous, this was further confirmed with MTT cytotoxicity assays. As in the case of **Ru1** and **RuL1**, both pyrene-ruthenium derivatives induced only a slight decrease of cell viability after 24 h of incubation at high concentrations (Supplementary Fig. 5).

**Intracellular reactivity of the pyrene-ruthenium derivatives**. The previous experiments confirmed that the complex **RuL2** has excellent cell membrane permeability, can be readily taken up by the cell mitochondria, is firmly retained inside the cell (even after a 20 min PBS washing, Supplementary Fig. 6) and does not compromise cell survival. A key question now was finding out whether the mitochondria-confined complex is catalytically active. Remarkably, the addition of **Rho-alloc** (100 µM) to cells that had been previously treated with TMRE (100 nM) and either ruthenium complex (50 µM), and thoroughly washed with DMEM-FBS, led to intracellular green staining patterns. Interestingly, while the green emission of the reaction product (**Rho**) observed in cells treated with **RuL2** matched almost

perfectly the red fluorescence of the TMRE dye as well as the blue fluorescence coming from **RuL2** (Fig. 4b, panels E and F, respectively and Fig. 5), in cells incubated with **RuL3** the green fluorescence of **Rho** showed a much poorer correlation with the mitochondrial dye TMRE (Fig. 4c, panel E). Particularly illustrative is the comparison between the overlays obtained from **RuL2** and **RuL3**, represented in panels E (Fig. 4b,c, respectively).

Also importantly, similar catalytic results, that could be fully replicated, were observed when the same experiments were carried out in adenocarcinomic human alveolar basal epithelial cells (A549 cells, Supplementary Fig. 9).

Using the Mander's method, the co-localization data were quantified and the results are displayed in Fig. 5 (ref. 40). As can be deduced from the values obtained, there is a notable difference between both catalysts. In cells incubated with **RuL2**, there is high co-localization of the mitochondrial marker TMRE with both the reaction product **Rho** and the catalyst, with values of 73% and 88%, respectively (red signal overlapping green and blue); however, the values drop to $\sim$35% in cells incubated with **RuL3**, which are similar to those observed for cells incubated with **Ru1**.

The above data are fully consistent with a pi-allyl-mediated organometallic reaction occurring in the mitochondria of living cells, reaction that generates products in a highly localized manner. Curiously, *in vitro* catalytic assays using cell lysates as reaction medium revealed that **RuL2** (5 mol%) had a very poor performance, as we only observed a 0.2% of conversion after 4 h at 37 °C (**RuL3** leads to a 12.4% conversion in the same conditions; Supplementary Table 1). Although this low activity is in great part due to solubility or aggregation problems, it comes to indicate that *in vitro* results do not necessarily correlate with the reactivity profile in the environment of living cells and/or specific organelles.

In consonance with the proposed mechanism of mitochondrial localization based on the membrane potential, we observed that adding a depolarizing agent-like carbonylcyanide *p*-trifluoromethoxy-phenylhydrazone (20 nM)[41] to cells pretreated with **RuL2** (50 µM) promotes a diffuse blue staining no longer associated to the mitochondria. As expected, addition of the pro-fluorescent probe **Rho-alloc** led to an accumulation of green fluorescence throughout the cytoplasm rather than in the mitochondria (Supplementary Fig. 10).

**Activation of a mitochondrial membrane-uncoupler agent**. While the above experiments demonstrated that the preferential accumulation of metal complex **RuL2** in mitochondria promoted a confined catalytic response, it remained to be shown whether such subcellular reactivity could provide a functional advantage. Towards this end, we paid attention to the possibility of a localized generation of 2,4-dinitrophenol (**DNP**), a protonophore known to decrease the mitochondrial membrane potential and switch off the ATP production (Fig. 6a)[42,43]. Importantly, control experiments confirmed that the allyl-caged derivative **DNP-allyl** failed to elicit any observable biological effect in the mitochondria, even at relatively high concentrations (500 µM). However, treatment of HeLa cells with 500 µM of the phenol-free form (**DNP**) produced a depolarization in the mitochondria, as can be deduced from the change in the fluorescence staining induced by the TMRE and its decrease in intensity (Fig. 6b, panel D)[44]. TMRE is a cell permeable, positively charged, red–orange dye that readily accumulates in active mitochondria. Depolarized or inactive mitochondria have decreased membrane potential and fail to sequester TMRE, which therefore generates a decreased intensity and a much less defined red staining (Supplementary Fig. 11).

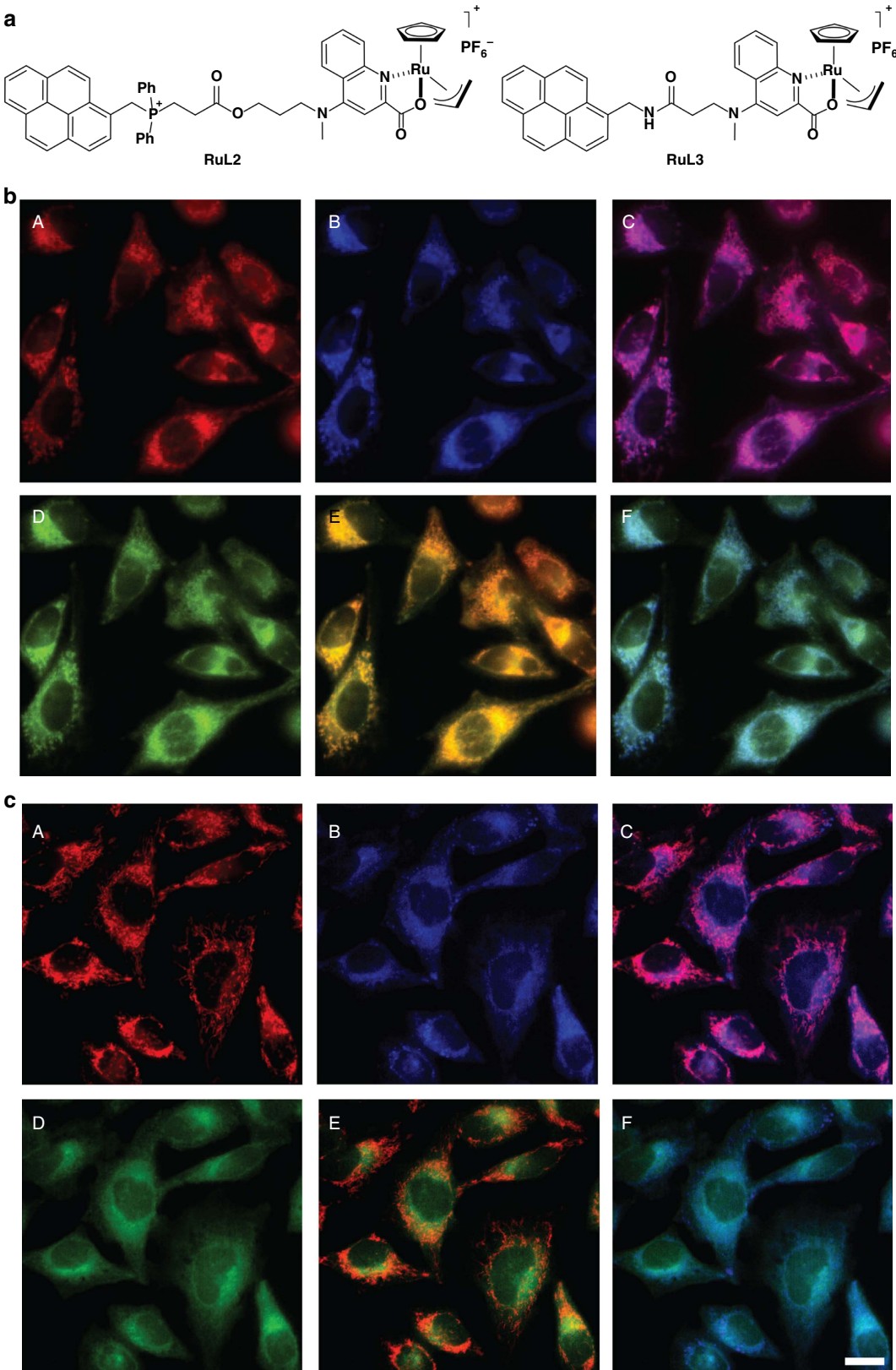

**Figure 4 | Reactivity of ruthenium complexes RuL2 and RuL3 in HeLa cells. (a)** Structure of the ruthenium complexes **RuL2** and **RuL3**. **(b)** Subcellular localization and catalytic activity of **RuL2** and **(c)** **RuL3**. (A) Mitochondrial labelling with TMRE (red), (B) emission of the metal complex (blue), (C) merging of A and B, (D) **Rho** fluorescence in cells preincubated with the ruthenium catalysts after addition of **Rho-alloc**, (E) merging of A and D, and (F) merging of B and D. Reaction conditions: cells were incubated with the catalyst (50 μM) and TMRE (100 nM) for 30 min, followed by two washings with DMEM-FBS and treatment with **Rho-alloc** (100 μM) for 30 min. Scale bar, 12.5 μm.

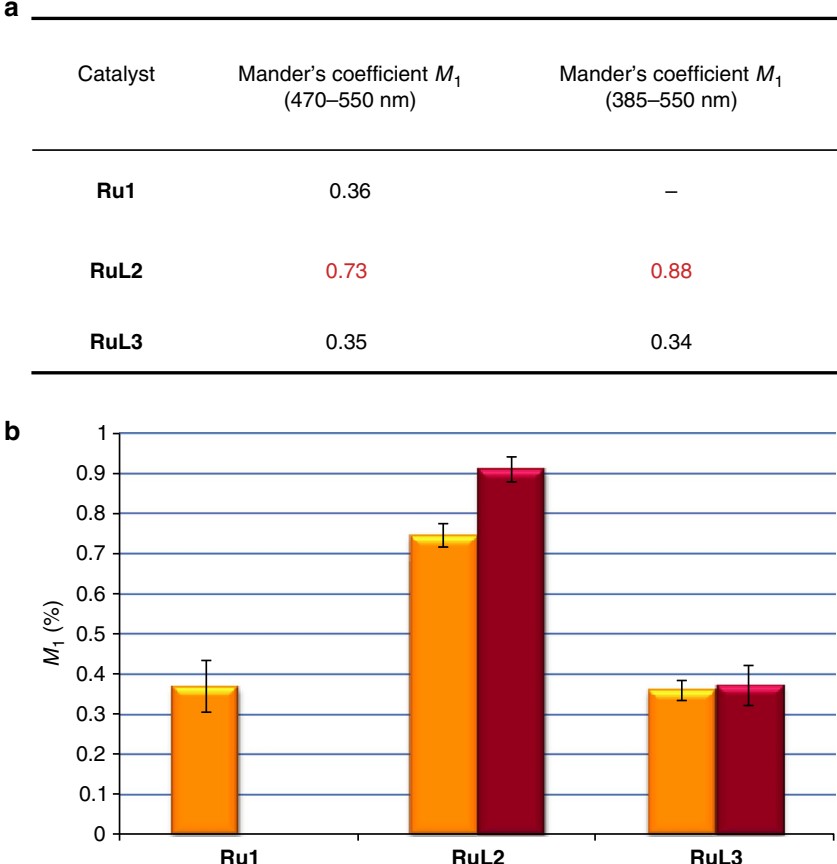

| Catalyst | Mander's coefficient $M_1$ (470–550 nm) | Mander's coefficient $M_1$ (385–550 nm) |
|---|---|---|
| **Ru1** | 0.36 | – |
| **RuL2** | 0.73 | 0.88 |
| **RuL3** | 0.35 | 0.34 |

**Figure 5 | Co-localization studies of the ruthenium catalysts in HeLa cells.** (**a**) Co-localization coefficients (Mander's coefficients, $M_1$) performed on dual-colour images from fluorescent microscopy experiments represented in Fig. 4. (**b**) Bars representation of the Mander's coefficients. Yellow bar: fraction of TMRE overlapping green signal of the reaction product **Rho**; magenta bar: fraction of TMRE overlapping blue signal of the catalyst. Error bars represent the standard deviation of measurements in three different samples.

Using TMRE as indicator, we found that neither metal complex **RuL2** nor **RuL3** produced major damages in the mitochondria potential, even after 24 h (Fig. 6b, panels A and B). However, the addition of **DNP-allyl** (150 μM) to the cells previously treated with **RuL2**, produced a rapid damage in the mitochondrial membrane potential, as deduced from the considerable decrease in the TMRE fluorescence observed after 30 min (Fig. 6b, panel C). This effect must arise from the subcellular confined release of the active uncoupler **DNP**, because addition of the same amount of substrate to cells pretreated with **RuL3**, which lacks the phosphonium-targeting moiety, produced a very weak depolarization effect (Fig. 6b, panel E). This result has been reproduced not only in HeLa but also in other mammalian cells such as Vero (Supplementary Fig. 12). Therefore, the mitochondria-confined catalyst led to a much more efficient and rapid response, most probably because of an intense, confined and probably amplified release of the damaging agent (Fig. 6c).

Also important, while the generation of a reproducible damaging effect with the active drug (**DNP**) required the use of concentrations > 500 μM, cells pretreated with **RuL2** were able to generate reproducible depolarization effects employing only 150 μM of the **DNP-allyl** precursor (Fig. 6d), which suggests that the confined catalyst not only enables the use of an inert prodrug but also allows to decrease the dosage.

In conclusion, we have described the first non-toxic, transition metal complex that is capable to promote an abiotic organometallic catalytic transformation of designed exogenous substrates within the mitochondria of living cells. Appropriate fluorescent labelling allows tracking the intracellular localization of the metal complexes, localization that depends on the structure of the metal ligands. The use of the pyrene-phosphonium-targeting group and a hydrophobic linker allows for a preferential mitochondrial accumulation of the ruthenium complex and ensures its intracellular retention. Importantly, we have also demonstrated that the confined installation of the metal catalyst can be used to obtain functional advantages, such as a smooth and rapid ruthenium-dependent depolarization of the mitochondria. These studies, by enabling catalytic functions that are absent in nature, open new avenues and opportunities in the use of typical transition metal catalysis for the selective manipulation of living systems.

## Methods

**General.** Chemical synthesis procedures, detailed protocols and characterization of all the compounds are included in the Supplementary Methods section. For NMR and high-resolution electrospray ionisation mass spectrometry (HR-ESI-MS)/ MALDI-TOF analysis of the compounds in this article, see Supplementary Figs 13–25.

**Synthesis of L1.** Allyl 4-[(3-hydroxypropyl)(methyl)amino]quinoline-2-carboxylate was synthesized from precursor 4-bromoquinoline-2-carboxylic acid (**1**)[33].

(3-Carboxypropyl)triphenylphosphonium bromide (46.0 mg, 0.11 mmol) was dissolved in *N,N*-dimethylformamide (DMF) (1 ml). 1-Ethyl-3-(3-dimethylaminopropyl)carbodiimide (52.0 mg, 0.27 mmol), allyl 4-[(3-hydroxypropyl)(methyl)amino]quinoline-2-carboxylate (32.0 mg, 0.11 mmol) and 4-dimethylaminopyridine (26.0 mg, 0.22 mmol) were subsequently added to the solution. The reaction mixture was left stirring overnight and the solvent was removed under reduced pressure. The crude was purified by flash chromatography on silica gel (DCM/MeOH 9:1) yielding an off yellow liquid (23.0 mg, 30%).

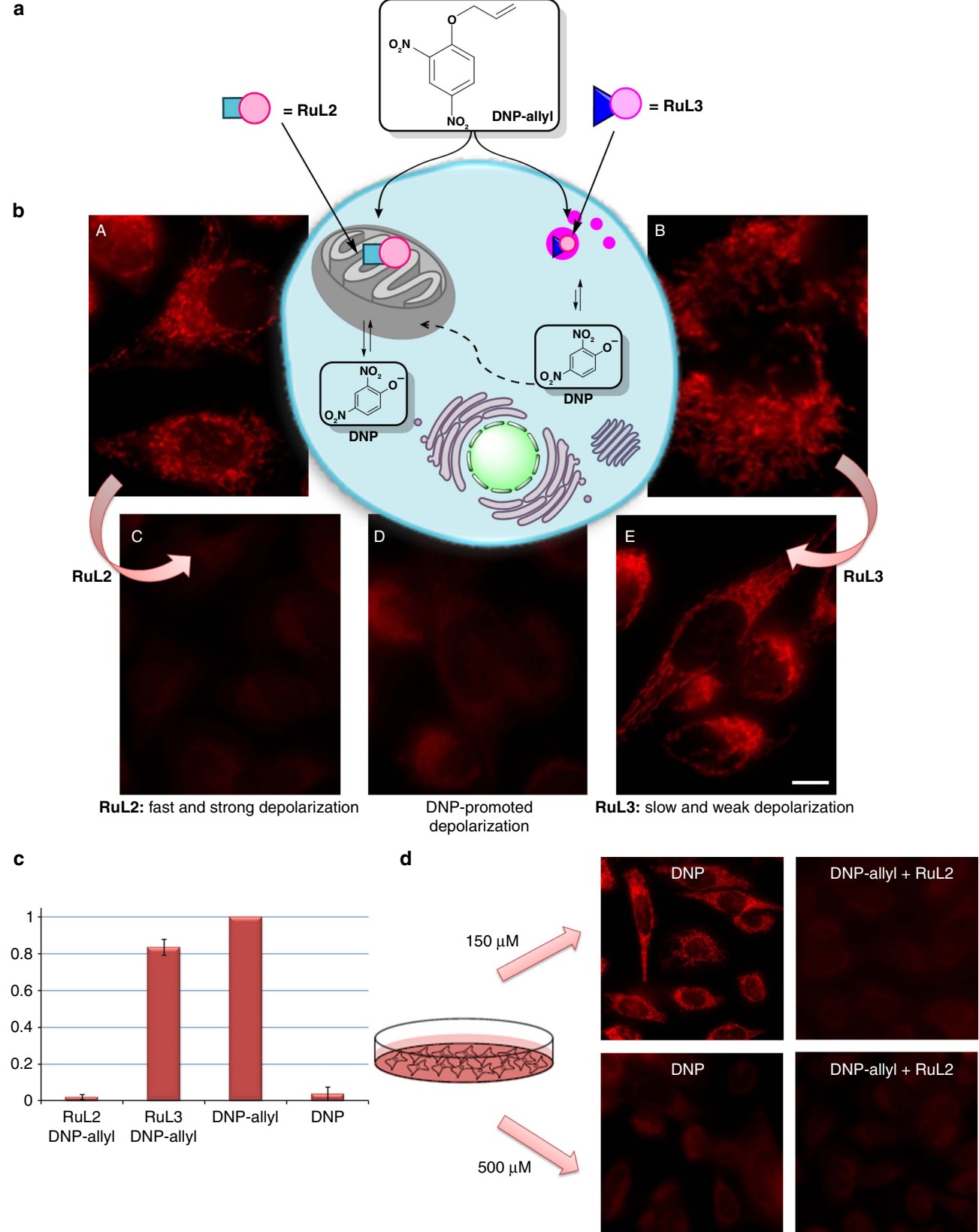

**Figure 6 | Activation of DNP-allyl inside HeLa cells.** (**a**) Schematic representation of the uncaging reaction of **DNP-allyl** with **RuL2** and **RuL3**. (**b**) Fluorescence micrographs of the mitochondrial staining with TMRE (100 nM) of: (A) cells incubated with **RuL2** (50 μM), (B) cells incubated with **RuL3** (50 μM), (C) cells treated with **DNP-allyl** (150 μM) after incubation with **RuL2**, (D) cells treated with **DNP** (500 μM) and (E) cells treated with **DNP-allyl** (150 μM) after incubation with **RuL3**. (**c**) Quantification of the fluorescence of the micrographs from section (**b**) showing the normalized decrease of the TMRE fluorescence. (**d**) Influence of the concentration of **DNP/DNP-allyl** on the mitochondrial depolarization of HeLa cells. Scale bar, 12.5 μm. Error bars represent the standard deviation of measurements in three different samples.

**Synthesis of the ruthenium complex RuL1.** To a 10 mM solution of ligand **L1** (1.0 equiv.) in anhydrous dichloromethane (DCM) was added 1 equiv. of [RuCp(CH$_3$CN)$_3$]PF$_6$. The reaction mixture was left stirring overnight at room temperature. The reaction mixture was centrifuged, the supernatant was collected and the solvent was removed under reduced pressure yielding the complex **RuL1** as a dark solid (65%).

**Cell culture experiments.** All cell lines were cultured in DMEM supplemented with 10% (v/v) FBS, 5 mM glutamine, penicillin (100 U ml$^{-1}$) and streptomycin (100 U ml$^{-1}$; all from Invitrogen). Proliferating cell cultures were maintained in a 5% CO$_2$ humidified incubator at 37 °C. Unless otherwise specified, all incubations were performed in DMEM supplemented with 5% of FBS (DMEM-FBS) at 37 °C.

**Viability assays.** The toxicity of the catalysts **Ru1** and **RuL1–RuL3** was tested by MTT assay in HeLa and A549 cell lines as follows: 15,000 cells per well were seeded in 96-well plates 1 day before treatment with different concentrations of the catalysts. After 24 h of incubation, medium containing Thiazolyl Blue Tetrazolium Bromide (Sigma) was added to a final concentration of 0.5 mg ml$^{-1}$. Cells were then incubated for 4 h to allow the formation of formazan precipitates by metabolically active cells. A detergent solution of 10% SDS and 0.01 M HCl was then added and the plate was incubated overnight at room temperature to allow the solubilization of the precipitates. The quantity of formazan in each well (directly proportional to the number of viable cells) was measured by recording changes in absorbance at 570 nm in a microtitre plate reading spectrophotometer (Tecan Infinite 200 PRO).

**Catalytic experiments in living cells.** *Uncaging of Rho-alloc.* Cells growing on glass coverslips were incubated with either catalyst **Ru1**, **RuL1**, **RuL2** or **RuL3** (50 μM) and TMRE (100 nM) for 30 min. Cells were then washed twice with DMEM-FBS and incubated with **Rho-alloc** (100 μM) for 30 min. Before observation by fluorescence microscopy, the samples were washed twice with fresh DMEM-FBS. The coverslips were observed *in vivo* in a fluorescence microscope equipped with adequate filters. Digital pictures of the different samples were taken under identical conditions of gain and exposure.

*Uncaging of DNP-allyl.* Cells growing on glass coverslips were incubated with either catalyst **RuL2** or **RuL3** (50 μM) for 30 min. Cells were then washed twice with DMEM-FBS and incubated with the caged **DNP-allyl** (500 μM) for 30 min. Finally, TMRE (100 nM) was added to the incubation medium for 10 min. The coverslips were then observed *in vivo* in a fluorescence microscope equipped with adequate filters. Digital pictures of the different samples were taken under identical conditions of gain and exposure.

**Data availability.** The authors declare that the data supporting the findings of this study are available within the article and its Supplementary Information files. Other data that support the findings of this study are available from the corresponding author on request.

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

## Acknowledgements

We are thankful for the support given by the Spanish grant SAF2013-41943-R, the Xunta de Galicia (GRC2013-041 and 2015-CP082), the ERDF and the European Research Council (Advanced Grant No. 340055). M.T.G. thanks the Ministerio de Economía y Competitividad for the Postdoctoral fellowship. M.M.C. thanks Ministerio de Economía y Competitividad for the Juan de la Cierva-Incorporación fellowship (IJCI-2014-19326). The authors thank R. Menaya-Vargas for excellent technical assistance and Professor P. Bermejo and Dr P. Herbello for their help with the ICP measurements, as well as Dr M.I. Sánchez for preliminary studies on the preparation of ligands.

## Author contributions

M.T.G., M.M.C. and J.R.C. conducted the experiments and analysed the experimental data. M.T.G., M.M.C., J.R.C. and J.L.M. co-wrote the paper. J.L.M. guided the research.

## Additional information

**Competing financial interests:** The authors declare no competing financial interests.

