## [Peer review file · Nature Communications]

Reviewers' comments:

Reviewer #1 (Remarks to the Author):

This communication by Mascarenas and coworkers describes the development of organometallic ruthenium complexes that are designed to accumulate in the mitochondria of living cells and which are capable to uncage alloc protected exogenous substrates in a bioorthogonal fashion. The authors give a comprehensive overview on the topic and cover the most important literature. They put their work into perspective to other biocompatible reactions mediated by metal ions (ruthenium, palladium) and enzymes (metalloenzymes). The authors then evaluated the catalytic activity of the synthesized metal complexes in solution and within living human cells with a caged rhodamine R110 derivative. They used a TPP delivery vector in the complex RuL1 to promote the accumulation of the active transition metal complex in the mitochondria and tested this accumulation with several co-staining experiments. The analogous catalyst RuL2, which itself is fluorescent, showed great colocalization with TMRE (a mitochondrial dye). Furthermore, the authors showed that the in cell uncaged rhodamine R110 accumulated as well within the mitochondria. When the depolarization agent FCCP was applied, a reduced mitochondrial accumulation of the catalyst was observed. In addition, the authors showed that the applied complexes present a low toxicity even after long (24h) incubation times. The authors applied the synthesized ruthenium complexes to activate the caged depolarization agent allyl-DNP within the mitochondria. Due to the confinement of the activation within the mitochondria, the effect of the active DNP compared to free DNP without any catalyst was increased.

This is exciting work on a topic that gains more and more attention, namely the design and application of synthetic catalysts for the controlled catalysis in living biological systems. To achieve this is an enormous challenge but promises to provide novel applications which exploit signal amplification due to catalyst turnover. The manuscript by Mascarenas builds on previous work by Meggers and also Wender but it is nevertheless very surprising since I would not have expected that the mitochondria-targeting works so beautifully and convincing! This work is therefore an important contribution to the area of bioorthogonal catalysis in living systems and will draw large attention. I therefore strongly support a publication of this work in Nature Communications.

The manuscript is written very clearly and prepared with the necessary attention to all details. However, the following two points should be addressed for the revisions:

1. The NMR spectra of the synthesized ruthenium complexes do not look really satisfactory. What is the reason for it? NMR measurements performed in the wrong solvent? Maybe acetone-d₆ would be more suitable. Furthermore, high resolution mass spectra are missing.
2. Ru1 was used as a reference compound which is not ideal since Meggers and coworkers demonstrated that electron donor substituents within the quinoline moiety have a significant effect on the catalytic properties. The authors should at least add a cautionary statement to the text.

Reviewer #2 (Remarks to the Author):

Overview

This is an elegant development that extends the well established methods to target active groups to mitochondria by conjugation to lipophilic cations. To the best of my knowledge, this is the first time that a transition metal catalyst has been so targeted. The data are nice proof of concept studies that open up a range of future directions.

Major points

1 For RuL1 and RuL2 the molecules are lipophilic dications, which should greatly enhance uptake into mitochondria, compared to a monocation. This should be discussed. Furthermore, the uptake of dications requires enhanced hydrophobicity to counteract the Born energy of the two charges in crossing a membrane. The authors have addressed this by the relatively hydrophobic linker, but the rationale for this should be discussed.

2 Ru1 is also a cation, but is presumably not taken up due to its relative hydrophilicity? This could be discussed.

3 It was a pity that the uptake and metabolism of RuL1 and RuL2 by isolated mitochondria was not explored. In particular, it would be interesting to assess the extent of uptake and also the rate at which the allyl anion was lost from the Ru(II). This is important as it was unclear if GSH in the cell would rapidly remove the allyl anion? If so, what would be the form that would cross the mitochondrial inner membrane? Would it cross as a dication with another ligand to partially counteract the Ru charge or as a trication, with the energetic problems that would entail? These issues should at least be discussed.

4 In Fig 3b was the greater activity of the cell lysates due to intact mitochondria?

5 Could the data in Fig 3d/e be explained by activation of rhodamine in the cytosol, followed by the well known uptake of rhodamine by energised mitochondria? Hence, this may not be proof that RuL1 is in the mitochondria in these cells.

6 The development of the phosphonium pyrene group is a nice and useful result!

7 The data in Fig 6 show nicely that there is activation of the DNP "pro-drug" and is a nice demonstration.

8 Were there any errors/statistical tests applied to the data in the bar charts and the tables?

Minor points

1 On page 5 the authors state: "...electrochemical gradient between the inner (IMM) and outer membranes." It would be more accurate to say that the accumulation is driven by the membrane potential across the mitochondrial inner membrane.

2 Personally, I don't like bar charts presented in 3D format as is done in Figs 5 and 6.

Author response to reviewer comments

1^o Referee:

1. The NMR spectra of the synthesized ruthenium complexes do not look really satisfactory. What is the reason for it? NMR measurements performed in the wrong solvent? Maybe acetone-d6 would be more suitable. Furthermore, high resolution mass spectra are missing.

We have tested the spectra in different solvents, and indeed found that in most of them the phosphonium containing complexes exhibit usually broad signals, most probable because of aggregation processes. Anyway we present considerably improved ¹H-NMR spectra carried out in CD₂Cl₂ (Supplementary Information Fig. 16-18).

Importantly, we provide mass spectra obtained using different techniques as well as high resolution mass spectra. All the data have been included in the supplementary information (Supplementary Information Fig. 19-21).

2. Ru1 was used as a reference compound which is not ideal since Meggers and coworkers demonstrated that electron donor substituents within the quinoline moiety have a significant effect on the catalytic properties. The authors should at least add a cautionary statement to the text.

Ru1 was used because it can be easily made, as the ligand precursor (quinoline 2-carboxylic acid) is commercially available. The main theme of the work is the study of localized catalysis; it's not intended to make a comparison of catalytic activities or kinetics, so the use of other more active quinoline derivatives as reference control was not needed.

We have added a comment in the text on that (marked in yellow)

2° Referee:

1. For RuL1 and RuL2 the molecules are lipophilic dications, which should greatly enhance uptake into mitochondria, compared to a monocation. This should be discussed. Furthermore, the uptake of dications requires enhanced hydrophobicity to counteract the Born energy of the two charges in crossing a membrane. The authors have addressed this by the relatively hydrophobic linker, but the rationale for this should be discussed.

We have introduced also a comment on the text of the main manuscript on these aspects (marked in yellow).

2 Ru1 is also a cation, but is presumably not taken up due to its relative hydrophilicity? This could be discussed.

The lipophilic character of the molecule is also crucial for the mitochondrial uptake. The hydrophobic aryl substituents of the phosphonium moieties are key for mitochondrial localization. **Ru1** lacks this unit, making its uptake more difficult and favouring its PBS washing (Supplementary Information Fig. 3). Our newly introduced ICP data (yellow marks) corroborated that those catalysts lacking the arylphosphonium moiety (Ru1 and RuL3) presented poor affinity for the mitochondria.

3 It was a pity that the uptake and metabolism of RuL1 and RuL2 by isolated mitochondria was not explored. In particular, it would be interesting to assess the extent of uptake and also the rate at which the allyl anion was lost from the Ru(II). This is important as it was unclear if GSH in the cell would rapidly remove the allyl anion? If so, what would be the form that would cross the mitochondrial inner membrane? Would it cross as a dication with another ligand to partially counteract the Ru charge or as a trication, with the energetic problems that would

entail? These issues should at least be discussed.

Well, we have indeed been able to develop experimental conditions to obtain isolated mitochondria from cells that had been treated with the ruthenium catalysts, and appropriately washed. These experiments confirmed the expected preferential accumulation in mitochondria: **RuL2**>>**RuL1**>**RuL3**~**Ru1**. All these data have been commented in the main text (yellow marks) and included in the supporting information (Supplementary Information Table 2).

Discovering the real complexation state of the ruthenium catalysts inside the cell is extremely difficult, and anyway, not so relevant in the context of our work.

4 In Fig 3b was the greater activity of the cell lysates due to intact mitochondria?

The lysis protocol includes a strong sonication step which probably disrupted the mitochondrial membranes. Therefore, we don't expect any specific association of RuL1 to mitochondria in the lysates.

A detailed protocol of how these cell lysates were obtained has been added to the supplementary information for clarification.

5 Could the data in Fig 3d/e be explained by activation of rhodamine in the cytosol, followed by the well-known uptake of rhodamine by energised mitochondria? Hence, this may not be proof that RuL1 is in the mitochondria in these cells.

The differences in the accumulation of the rhodamine in the mitochondria with the different catalyst employed cannot be explained only with the migration of the product towards this organelle. Indeed, in the case of Ru1 the amount of rhodamine product in the mitochondria is relatively low, which suggests that once it's formed it doesn't migrate substantially during the time of the experiment. On the other hand, the ICP experiments correlate very well the presence of Ru in the mitochondria with the amount of Rho generated in this organelle.

6 The development of the phosphonium pyrene group is a nice and useful result!

7 The data in Fig 6 show nicely that there is activation of the DNP "pro-drug" and is a nice demonstration.

8 Were there any errors/statistical tests applied to the data in the bar charts and the tables?

The corresponding error bars have been implemented in graphics of Fig. 3b and 3c, Fig. 5b and Fig. 6c.

Minor points

1 On page 5 the authors state: "...electrochemical gradient between the inner (IMM) and outer membranes." It would be more accurate to say that the accumulation is driven by the membrane potential across the mitochondrial inner membrane.

The text has been changed accordingly.

2 Personally, I don't like bar charts presented in 3D format as is done in Figs 5 and 6. The format of graphics of Fig. 5 and 6 have been changed.

REVIEWERS' COMMENTS:

Reviewer #1 (Remarks to the Author):

My high enthusiasm for the manuscript by Mascarenas and coworkers is unchanged. I strongly support a publication of this work in Nature Communications.

The authors addressed all my points of critique in a satisfactory fashion.

Reviewer #2 (Remarks to the Author):

none